# In or Out? New Insights on Exon Recognition through Splice-Site Interdependency

**DOI:** 10.3390/ijms21072300

**Published:** 2020-03-26

**Authors:** Mubeen Khan, Stéphanie S. Cornelis, Riccardo Sangermano, Iris J.M. Post, Amber Janssen Groesbeek, Jan Amsu, Christian Gilissen, Alejandro Garanto, Rob W.J. Collin, Frans P.M. Cremers

**Affiliations:** 1Department of Human Genetics, Radboud University Medical Center, 6525 GA Nijmegen, The Netherlands; mubeen.khan@radboudumc.nl (M.K.); stephanie.cornelis@radboudumc.nl (S.S.C.); Riccardo_Sangermano@MEEI.HARVARD.EDU (R.S.); amber.janssengroesbeek@radboudumc.nl (A.J.G.); jan_amsu@hotmail.com (J.A.); christian.gilissen@radboudumc.nl (C.G.); alex.garanto@radboudumc.nl (A.G.);; 2Donders Institute for Brain, Cognition and Behaviour, Radboud University Medical Center, 6525 GA Nijmegen, The Netherlands; 3Ocular Genomics Institute, Department of Ophthalmology, Massachusetts Eye and Ear Infirmary, Harvard Medical School, Boston, MA 02114, USA; 4Radboud Institute for Molecular Life Sciences, Radboud University Medical Center, 6525 GA Nijmegen, The Netherlands

**Keywords:** Pre-mRNA, splicing, 5′ and 3′ splice sites, interdependency

## Abstract

Noncanonical splice-site mutations are an important cause of inherited diseases. Based on in vitro and stem-cell-based studies, some splice-site variants show a stronger splice defect than expected based on their predicted effects, suggesting that other sequence motifs influence the outcome. We investigated whether splice defects due to human-inherited-disease-associated variants in noncanonical splice-site sequences in *ABCA4*, *DMD*, and *TMC1* could be rescued by strengthening the splice site on the other side of the exon. Noncanonical 5′- and 3′-splice-site variants were selected. Rescue variants were introduced based on an increase in predicted splice-site strength, and the effects of these variants were analyzed using in vitro splice assays in HEK293T cells. Exon skipping due to five variants in noncanonical splice sites of exons in *ABCA4*, *DMD*, and *TMC1* could be partially or completely rescued by increasing the predicted strengths of the other splice site of the same exon. We named this mechanism “splicing interdependency”, and it is likely based on exon recognition by splicing machinery. Awareness of this interdependency is of importance in the classification of noncanonical splice-site variants associated with disease and may open new opportunities for treatments.

## 1. Introduction

Pre-mRNA splicing of multi-exon genes is performed by the spliceosome, which recognizes specific sequence motifs at exon boundaries [1,2,3]. Variants in these motifs can change the binding strength of the spliceosome, which can alter splicing [4]. This can result in disease when necessary (parts of) exons are skipped or (parts of) introns are retained, causing frameshifts, premature stop codons, or disruptions in protein folding [5].

In order to predict the effect of genetic variants on splicing, many splice-site prediction programs have been created. Five splicing prediction programs are used in the commonly used program Alamut Visual (Interactive Biosoftware, Rouen, France, version 2.7) to increase the reliability of this in silico analysis. Splicing prediction scores are provided for both the reference and alternative sequence, not only for the splice sites, indicating the known exon–intron boundaries but also alternative (cryptic) splice sites in nearby sequences. Decreased splicing scores often result in exon skipping, but exon elongation and partial exon truncation are also observed when alternative splice sites are recognized by the splicing machinery [6,7,8]. Different algorithms are used in various splicing prediction programs. SpliceSiteFinder-like [9,10], MaxEntScan [11], NNSPLICE [12], GeneSplicer [13], and Human Splicing Finder [14] are incorporated into the Alamut software and take into account a maximum of 80 nucleotides on either side of the variant. The exons in human DNA are relatively small (average 145–150 bp; mean 120 bp) [15,16] and introns are generally much larger (average 5.5 kb; mean 3.3 kb) [17,18], which likely forms the basis for an initial “exon recognition” by the spliceosome. Surprisingly, the above-mentioned splicing algorithms assess the strengths of the 5′ and 3′ splice sites independently, not taking into account that exon recognition may also depend on the strength of the splice site at the other side of an exon.

In 2002, Hefferon et al. showed how exon skipping due to a short polypyrimidine tract upstream of *CFTR* Exon 9 at the 3′ splice site could be rescued by an adenine insertion between positions +3 and +4, strengthening the 5′ splice site of Exon 9. In vivo studies of mice and sheep supported this finding [19]. Later it was found that natural skipping of Exon 6 of the *CHRNE* gene could be reduced by improving either the 5′ or 3′ splice site of Exon 6 by changing at least four nucleotides in the sequence [20], indicating that if one of the splice sites of *CHRNE* Exon 6 is weak and the other is strong, normal splicing can still take place.

By studying the effect of a predicted very mild 3′-splice-site variant (c.5461-10T>C) in *ABCA4* (NM_000350.2) in individuals with Stargardt disease (STGD1), we observed a very strong splice defect (Exon 39 or Exon 39 and 40 skipping) in mRNA isolated from patient-derived photoreceptor precursor cells [21]. We hypothesize that the large effect of the very mild 3′-splice-site variant was due to a weak 5′ splice site of Exon 39.

Here, we aimed to study whether splice-site interdependency represents a more general phenomenon. Next to the noncanonical splice-site (NCSS) variant c.5461-10T>C in *ABCA4*, we selected additional predicted weak 3′ NCSS variants in introns of *DMD* (NM_004006.2) and *TMC1* (NM_138691.2), which were accompanied by relatively weak 5′ splice sites [22,23]. Each of them has been previously shown to result in exon skipping and was associated with inherited human disease. Employing in vitro minigene splice assays to strengthen the 5′ splice site by changing their sequence towards the 5′ NCSS consensus sequence [24] enabled us to study the effect on splicing of the concerned exon as well as adjacent exons. Similarly, we investigated whether the exon skipping of two *ABCA4* exons due to NCSS variants at the 5′ splice site could be corrected by strengthening the 3′ splice site, to investigate whether the dependency between splice sites works both ways.

## 2. Results

### 2.1. Rescuing the Exon-Skipping Effect of 3′-Splice-Site Variants

The 3′-splice-site variant c.5461-10T>C constitutes the most frequent severe *ABCA4* variant in STGD1. When present alone in a minigene splice construct and transfected into Human Embryonic Kidney 293T (HEK293T) cells, it resulted in different Exon 39/40 skipping RNA products (Figure 1). Densitometric analysis showed that RNA transcribed from the wild-type (WT) construct also showed a fair amount of exon skipping (1% of Exon 39 skipping and 31% of Exon 39–40 skipping, Figure 1C, Appendix A), also shown previously [21]. The BA26 minigene construct carrying c.5461-10T>C was also mutagenized to contain the alternative nucleotides A, C, or T at Position c.5584+4. Based on Human Splicing Finder (HSF), the +4A variant strengthened this 5’ splice site significantly (Figure 1E; from 83.8 to 92.1). The +4C and +4T variants, also according to HSF, did not strengthen this splice site, but were inserted to investigate another hypothesis. The +4 to +6 nucleotides were G’GT, which were part of a weak predicted 5′ splice site (HSF: 69.0) which could negatively influence proper Exon 39 recognition. The +4G>C and +4G>T changes could potentially remove this effect and thereby result in less Exon 39 skipping in the presence of the WT 3′-splice-site sequence c.5461-10T, or the mutant variant c.5461-10C. The c.5584+4G>A variant, but not c.5584+4G>C or c.5584+4G>T, clearly rescued the splicing defect due to c.5461-10T>C (Figure 1D–F, Appendix A). Interestingly, the c.[5461-10T>C;5584+G>A] construct showed more correctly spliced mRNA than the WT construct. However, compared to the WT mRNA, a new band of ~400 nucleotides was present too, but this band could not be sequenced, which suggested that it was probably a heteroduplex fragment. As shown in Figure 1A,D, the +4G>C and +4G>T changes did not result in a clear rescue of erroneous splicing in the presence of the WT or mutant 3′ splice site, respectively, suggesting that the G’GT sequence at Positions +4 to +6 did not significantly affect the strength of the 5′ splice site. Thereby, we showed that exon skipping due to a weak variant in the 3′ splice site of *ABCA4* Exon 39 could be fully rescued by strengthening the 5′ splice site of the same exon.

The next variant we investigated for splice-site interdependency was the pathogenic NCSS mutation c.1705-5T>G at the 3′ splice site of *DMD* Exon 15, which is associated with Becker muscular dystrophy (www.lovd.nl/DMD) [23]. Minigenes of *DMD* Exons 14–16 were used to examine the effect of this mutation and a putative rescuing variant c.1812+4T>A on the splicing of these exons. Variant c.1705-5T>G led to skipping of Exon 15 in the minigene construct, as shown by RT-PCR (Figure 2A–C, Appendix A). The exon skipping was largely rescued by introducing the c.1812+4T>A variant at the 5′ splice site downstream of the exon in a construct containing c.[1705-5T>G;1812+4T>A]. The densitometric analysis of RT-PCR products showed a rescue of ~73% of WT splicing as a result of the construct containing the splice-site variant and the rescue variant (Figure 2C, Appendix A). Therefore, the partial rescuing effect of increasing a 5′ splice site of an exon with a weak 3′ splice site is also possible in the human *DMD* gene.

The third pathogenic NCSS variant that we investigated was c.237-6T>G at the 3′ splice site of Exon 8 in *TMC1*, which has been associated with autosomal recessive hearing impairment. Minigenes of *TMC1* containing Exons 7 and 8 were generated with either the WT sequence, the mutation alone, one of the potential rescuing variants alone (i.e., c.362+4T>A or c.362+6G>T), or combinations of the mutation with the potential rescuing variants. RT-PCR of the WT construct led to a fragment of 459 bp containing *TMC1* Exons 7 and 8 as well as *RHO* Exon 5 (Figure 2D, Appendix A). The c.237-6T>G variant alone led to skipping of Exon 8. The c.[237-6T>G;362+4T>A] and the c.[237-6T>G;362+6G>T] constructs both resulted in WT mRNA, but also in some skipping of Exon 8. They also resulted in an additional sequence (marked with an asterisk in Figure 2D), which turned out to be a heteroduplex fragment. Furthermore, all the constructs apart from the mutant construct led to a small fraction of a specific product (marked with a hashtag) that mapped to Chromosome 17. The c.[237-6T>G;362+4T>A] construct led to more WT product (52%) than the c.[237-6T>G;362+6G>T] construct (29.5%) (Figure 2F, Appendix A). This was not unexpected as, based on the human splice-site consensus sequence, the 5′ splice site +4A nucleotide is more prevalent than the +6T nucleotide [24]. Altogether, we showed that 3′ NCSS intronic variants that decrease or completely disrupt splicing can be partially or fully rescued by strengthening the noncanonical intronic region of the 5′ splice site in multiple genes.

### 2.2. Rescuing the Exon-Skipping Effect of NCSS Variants at the 5′ Splice Site

After showing that NCSS intronic variants at the 3′ splice site of exons in several genes could be rescued by strengthening the 5′ splice site in vitro, we investigated whether pathogenic NCSS variants at the 5′ splice site of exons in the *ABCA4* gene might be rescued similarly by introducing a NCSS variant at the 3′ splice site of the same exon. The first variant we analyzed was the intronic *ABCA4* c.302+4A>C variant downstream of Exon 3, which led to complete skipping of Exon 3 in the BA1 construct. Introducing the c.161-3A>C variant at the 3′ splice site of the exon in the mutant construct completely rescued the exon-skipping to WT ratios (Figure 3A–C, Appendix A).

The exonic *ABCA4* variant c.6478A>G residing at the last nucleotide position of Exon 47, which has been shown to diminish correct splicing of Exon 47 by almost 50%, was previously introduced into minigene BA29 [25]. Mutagenizing the cytosine into a guanine at Position c.6387 at the 5′ end of Exon 47 fully rescued the exon-skipping effect of variant c.6478A>G (Figure 3D–F, Appendix A). This also indicated that weak 5′ splice sites can be rescued by strengthening the 3′ splice site of the same exon.

## 3. Discussion

In this study, we showed that for exons of different genes, exon skipping due to mutations of the NCSS sequences of either the 5′ or the 3′ splice site can be undone by strengthening the 5′ or 3′ splice sites, respectively. This phenomenon, which we named “splicing interdependency”, can be widespread, which warrants the development of new splice-site-strength algorithms that take into account the strengths of both splice sites of exons. Jaganathan et al. recently reported on a new splice-site prediction tool, called “SpliceAI”, which is based on a larger sequence context of an investigated variant [26]. SpliceAI provides splice-site predictions for 5′ and 3′ splice sites of exons and pseudoexons, and thereby confirms that our findings are relevant for all multi-exon genes.

In Figure 4, we provide a model for splicing interdependency. According to our model, a NCSS variant results in less splicing factors (SFs) binding to one end of the exon (Figure 4A–C) in pre-mRNA, which has an effect on the exon recognition by the splicing machinery, as the “cooperation” between SFs at the exon boundaries is lost (compare Figure 4B,C). Strengthening the opposite splice site (Figure 4D) recruits extra SFs and thereby compensates for the loss of SFs binding at the NCSS variant site.

Previously, Zeniou et al. tried to rescue exon skipping in a similar way by increasing the polypyrimidine stretch with a TCTC insertion in the 3′ splice site of Exon 6 of *RPS6KA3*, formerly known as *RSK2*, when the 5′ splice site of that exon was known to be weakened due to a +3A>G change [27]. The exact location of the TCTC insertion was not mentioned, but a TCTC insertion on the logical positions (-3_-2ins, -5_-4ins or -9_-8ins) to improve the polypyrimidine stretch, did not lead to an increased 3′-splice-site score according to any of the prediction programs in Alamut Visual, which might explain why the exon skipping was not rescued. As mentioned before, Hefferon et al. and Ohno et al. showed that aberrant splicing of the *CFTR* gene or reduced splicing of the *CHRNE* gene could be rescued by strengthening the alternative splice sites [19,20]. Similarly, our results showed that this is a mechanism common to multiple genes. Scalet et al. (2017) showed that strengthening of the 3′ or 5′ splice sites of *ATR* Exon 9 could partially or fully rescue, respectively, the effect of an exonic variant that introduced a cryptic 5′ splice site [28].

Our data clearly showed that a splice site’s strength is not solely dependent on its nearby sequence context—for which splicing prediction programs usually take into account fewer than 100 nucleotides—but also on the broader context of the pre-mRNA. It is likely that because the splicing machinery requires both a 5′ and 3′ splice site that flank an intron, the loss of a 3′ splice site makes the upstream 5′ splice site competes with the 5′ splice site of the affected exon to splice together with the next downstream 3′ splice site. This results in exon-skipping of the affected exon. Increasing the strength of the 5′ splice site of the affected exon will increase its chance to bind to the downstream 3′ splice site. The 5′ splice site of the exon upstream of the affected exon is then forced to splice with the weakened 3′ splice site by the lack of a stronger 3′ splice site.

Furthermore, another contextual factor that might affect the splicing efficiency of splice sites is the length of the adjacent exon and introns. Here, we investigated the effect of NCSS mutations of exons with lengths between ~100–130 bp between introns of ~100–8000 bp. Our results fit the exon definition model of Susan Berget, in which splice sites of one exon communicate with each other and thus depend on each other [29]. However, as it is possible that intron definition may also occur in the human genome [30], exons and introns that are respectively bigger or smaller than those investigated here might show different splice-site interdependency and thereby a different mechanism underlying exon skipping. In particular, larger exons might be more subject to flanking intron definition versus exon definition, which might alter the mode of splicing. Interestingly, intron length is not related to the speed of intron removal in humans [31], although this is not the case in flies [32]. This could mean that the intron length has no or little effect on splice-site recognition in humans.

In this study, predicted exon and intron splice enhancers and silencers were not taken into account. However, prediction scores have been consulted (Appendix A). The importance of also considering exonic splicing enhancers (ESEs) was illustrated by a study from Jin et al. [33]. They showed in a minigene construct containing the 134 bp long Exon 9 of the *ITGB3* gene (a.k.a. *GPIIIa*) that a C>A polymorphism 13 nt downstream of the Exon 9 3′ splice site (c.1138) or a G>A mutation 6 bp upstream of the Exon 9 5′ splice site (c.1255) did not lead to exon skipping, while a combination of these exonic variants did result in exon skipping [33]. Splicing predictions of the canonical splice sites only showed a slight decrease for the 3′ splice site due to c.1138C>A, as predicted by GeneSplicer. In contrast, ESE predictions showed that both the former and the latter variants disrupted more strong ESEs than they created, while the ESS predictions show that ESSs of the c.1138A variant slightly increased in total strength. This, as well as other studies [34,35], suggests that splicing interdependency does not only depend on the 5′ and 3′ splice sites, but also on ESE and ESS motifs.

Additional studies are needed to design an exon-skipping-prediction program, which will be important for the identification of disease-causing variants by taking into account both involved splice-site strengths and ESE and ESS motifs.

Furthermore, future studies will be needed to also investigate the effect of adjacent splice sites on exon skipping and to determine whether these splice sites should also be considered in exon skipping predictions.

Finally, our findings are of therapeutic importance. We previously found significant natural exon-skipping events leading to frameshifts and likely mRNA degradation for *ABCA4* Exons 3 and Exons 39/40 [25]. For both, we showed correction of NCSS variant-induced exon skipping by increasing the strength of the other splice site. This means that RNA splicing modulation can potentially be applied to motifs that influence the functionality of either the 5′ and 3′ splice site, irrespective of which of the two is mutated in its NCSS sequences. A prime example for the therapeutic potential is the steric hindrance of an intronic *SMN2* silencer element using an antisense oligonucleotide that boosts the inclusion of Exon 7 in *SMN2* mRNA and effectively treats infantile-onset spinal muscular atrophy [36,37]. Similar strategies can now be employed for a wide range of inherited diseases that are due to erroneous RNA splicing.

In conclusion, the exon-skipping effects of several NCSS variants associated with inherited diseases could be corrected by increasing the strength of the other splice site of the affected exon. This shows that exon inclusion strongly depends on the combined strengths of its 5′ and 3′ splice sites. However, most of the utilized splice-site-prediction programs do not incorporate the strength of the opposite splice site, although one recently developed program (SpliceAI) does. In addition, these findings have important repercussions for the design of RNA-splicing-modulation therapeutics, since RNA-binding molecules designed to promote exon inclusion or pseudoexon exclusion can now be designed for much broader target sequences.

## 4. Materials and Methods

### 4.1. Noncanonical Splice-Site Variant Selection

To assess the role of 5′- and 3′-splice-site-strength interdependency in the proper splicing of exons, NCSS variants in inherited-disease-associated genes were selected based on the following criteria: (i) found at least once in patients (www.lovd.nl), (ii) present in a gene frequently mutated in an autosomal recessive inherited disease, (iii) leading to a reduction in the strength of the splice site as calculated by at least one of the five algorithms (SpliceSiteFinder-like, MaxEntScan, NNSPLICE, GeneSplicer, Human Splicing Finder) [10,11,12,13,14] via Alamut Visual software version 2.7 (Interactive Biosoftware; Rouen, France; www.interactive-biosoftware.com), and (iv) the other splice site of the affected exon had a score of 80 or lower from SpliceSiteFinder-like and a score of 90 or lower by Human Splicing Finder. NCSS variants located at the 3′ splice site of the genes *TMC1* (c.237-6T>G), *DMD* (c.1705-5T>G), and *ABCA4* (c.5461-10T>C), and NCSS variants located at the 5′ splice site of two exons of the *ABCA4* gene (c.302+4A>C and c.6478A>G) were selected for further analysis [21,22,23].

### 4.2. Selection of 5′-Splice-Site Rescue Variants

Downstream of the 5′ splice site position c.5584 (HSF: 75.8) in *ABCA4*, there is a GGT sequence at c.5584+4G_5584+6T, which forms a cryptic splice site (HSF: 69.0 at +5) and might diminish the correct splicing of Exon 39. Therefore, the position c.5584+4G was selected to be mutagenized to C, A, or T in both the WT and the 3′-splice-site mutant (c.5461-10T>C) constructs to assess their effect on splicing. In this way, the cryptic splice site decreased in splicing strength prediction and the canonical splice site either increased or decreased in splicing-strength prediction, as shown by Alamut Visual (Appendix A). We hypothesized that a decreased splicing prediction of the canonical splice site might still lead to increased correct splicing due to the loss of the cryptic splice site.

To increase the overall strength of the 5′ splice sites of the two remaining selected variants, noncanonical 5′ splice-site positions between +4 and +10 were assessed for their effect on the splice-site scores. Based on the splice-site consensus sequence [24], certain nucleotides are more conserved than others; e.g., at Position +4, an adenine is more prevalent than other nucleotides. Therefore, in *DMD*, the 5′-splice-site sequence c.1812+4T (HSF: 73.7) was selected to be mutagenized into c.1812+4A (HSF 82.5) in both WT and 3′-splice-site mutant (c.1705-5T>G) constructs to increase the 5′-splice-site strength (Appendix A).

Similarly, to decrease the effect of the *TMC1* 3′-splice-site mutant (c.237-6T>G), the 5′ splice site position c.362+4T (HSF: 78.7) was selected to be mutagenized into c.362+4A (HSF: 82.5) (Appendix A). This strengthened a cryptic splice site at +6. Therefore, we also mutagenized position c.362+6G into c.362+6T, both separately and in combination (c.[362+4T>A;362+6G>T]), leading to three different rescue constructs.

### 4.3. Selection of 3′-Splice-Site Rescue Variants

Two NCSS 5′-splice-site variants in the *ABCA4* gene that are known to cause exon skipping [25] were selected to investigate whether the skipping effect of these types of mutations could also be rescued by strengthening the upstream 3′ splice site of the relevant exon. The first variant was c.302+4A>C, which was shown to cause complete skipping of Exon 3 in the BA1 construct. The c.161-3A>C variant was introduced into the mutant construct to increase the 3′-splice-site strength (HSF: 82.6 to 92.0) (Appendix A). The second variant was c.6478A>G, causing partial Exon 47 skipping in the BA29 construct. The variant c.6387C>G (HSF: 87.4 to 91.6) (Appendix A) was also introduced into the mutant construct, resulting in c.[6387C>G;6478A>G].

### 4.4. Generation of Wild-Type Minigenes

To generate the WT minigene constructs for each gene (*ABCA4, DMD*, and *TMC1*) bacterial artificial chromosome (BAC) clones were used. BACs were obtained from the BACPAC Resource Center (BPRC) at the Children’s Hospital Oakland Research Institute (Oakland, CA, USA). Using NucleoBond Xtra Midi EF (cat. no. 740420.250, Macherey-Nagel GmbH & Co. KG, Düren, Germany), BAC DNA was isolated and used as a template to generate large WT fragments for each gene, such that the fragments contained multiple exons when possible. For *TMC1*, a region of 7.2 kb containing Exons 7 and 8, and for *DMD*, a 9.4 kb region containing Exons 14, 15, and 16 were amplified using Gateway-tagged PCR primers (Appendix A) designed as described elsewhere [25]. Inclusion of Exon 9 into the *TMC1* construct was technically impossible, as it lies 9.5 kb downstream of Exon 8. Together with other sequences, this would have exceeded the maximal insert size (11.7 kb) that we were able to clone in the Gateway system at present [25]. For *ABCA4*, the WT constructs BA1 (Exon 1–3), BA26 (Exon 38–41) and BA29 (Exon 46–48) were used [25]. Purification of all the amplified fragments was performed using a PCR cleanup kit (cat. no.740609.250, Macherey-Nagel, GmbH & Co. KG, Düren, Germany) according to the manufacturer’s protocol prior to Gateway cloning, which was performed as described by Sangermano et al. 2018. All the WT constructs had been previously Sanger sequenced [25].

### 4.5. Mutant Minigene Generation

To introduce the variants at the 3′ splice sites of the WT constructs of the respective genes, site-directed mutagenesis primers were designed by using the online quick primer tool (http://www.genomics.agilent.com/primerDesignProgram.jsp?toggle=uploaNow&mutate=true&_requestid=1039517, accessed 10 July 2017). For the generation of double mutants, 3′-splice-site mutant constructs and WT constructs were used as a template to introduce the rescuing variant at the selected 5′-splice-site positions. Mutagenesis PCR was performed as described elsewhere [25]. Presence of the introduced mutation was confirmed by Sanger sequencing. Details of the site-directed mutagenesis primers and Sanger sequencing validation primers are provided in Appendix A.

### 4.6. In Vitro Splice Assay and RT-PCR Assessment

WT and mutant constructs for *ABCA4*, *DMD*, and *TMC1* were transfected in HEK293T cells, and the extracted total RNA was subjected to reverse transcription (RT)-PCR as previously described [25].

### 4.7. Quantification of RT-PCR Products

To assess the quantity of the mis-spliced and correctly spliced RT-PCR products, densitometric analysis was performed using Image J software after gel electrophoresis as described elsewhere [38].

## Figures and Tables

**Figure 1 ijms-21-02300-f001:**
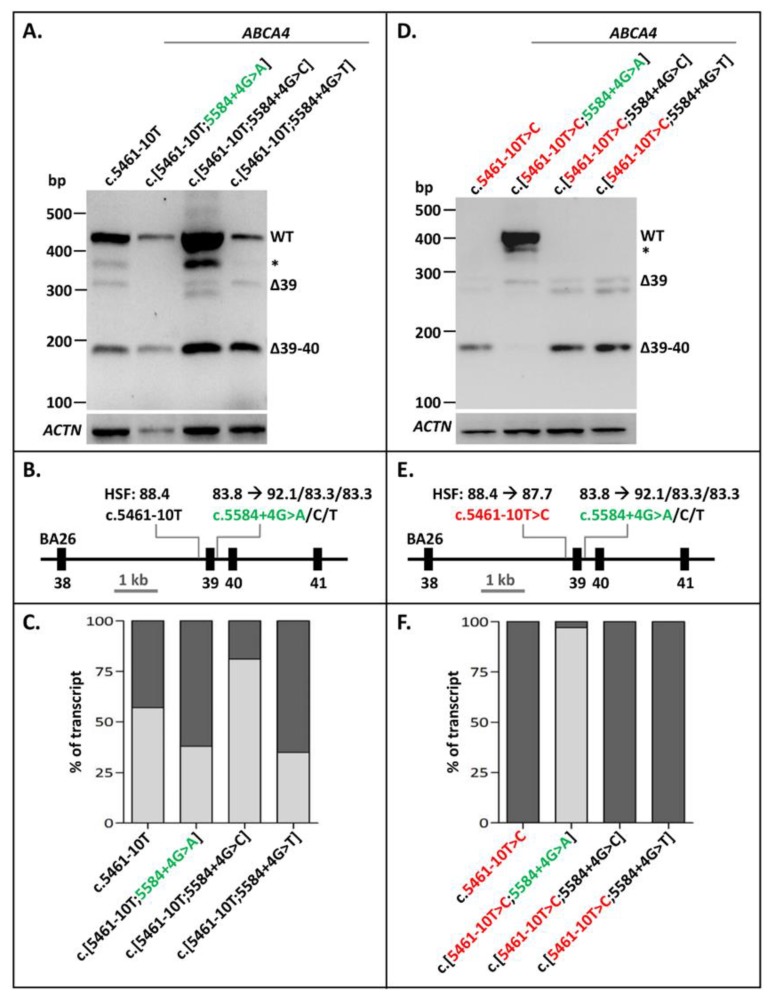
Rescue of Exon 39/40 skipping due to *ABCA4* variant c.5461-10T>C by 5′-splice-site strengthening. (**A**) The effect of altered nucleotides at Position c.5584+4 in the WT BA26 construct. None of the variants had a significant effect on Exon 39/40 splicing. Natural skipping of Exons 39/40 was observed for all. (**B**) Schematic overview of the BA26 construct used in A annotated with the Human Splicing Finder (HSF) scores with a range of [0–100], of which a higher score indicates a stronger splice prediction. (**C**) Semi-quantification of the ratio of correctly (light gray rectangles) and aberrantly spliced (dark gray rectangles) RT-PCR products due to altered nucleotides at Position c.5584+4 in the WT BA26 constructs. When multiple aberrant products were observed, the percentages were summed up. (**D**) The effects of altered nucleotides at c.5584+4 in the BA26 construct containing the c.5461-10T>C variant were that c.5584+4G>A, but not +4G>C or +4G>T, rescued the exon skipping due to c.5461-10T>C. (**E**) Schematic overview of the BA26 construct used in B. (**F**) Semi-quantification of the ratio of correctly (light gray rectangles) and aberrantly spliced (dark gray rectangles) RT-PCR products in BA26 construct containing c.5461-10T>C alone and together with rescue variants. * Band was identified as heteroduplex by Sanger sequencing. Red lettering indicates pathogenic variant. Green lettering indicates variant that rescued exon skipping.

**Figure 2 ijms-21-02300-f002:**
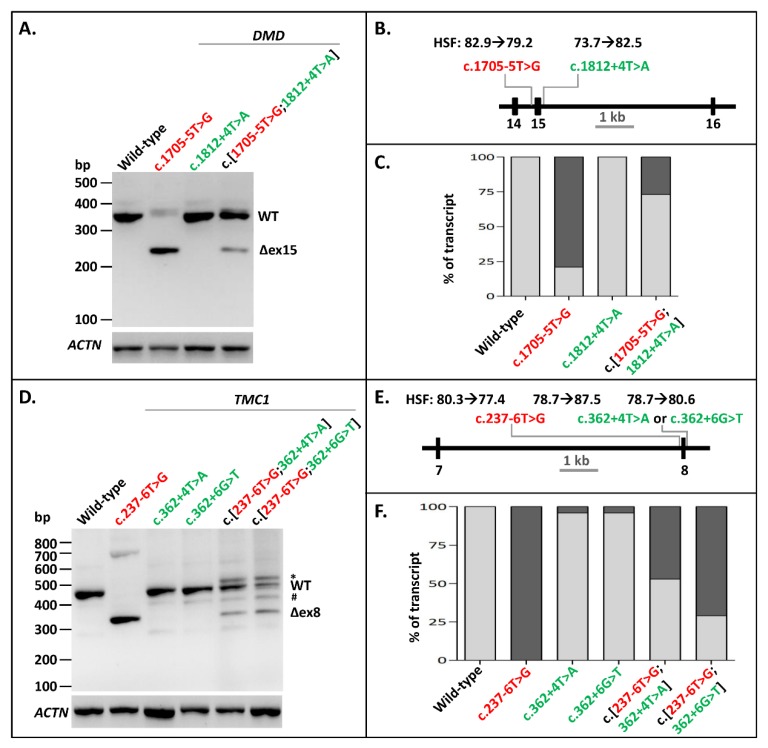
Splice defect rescue of pathogenic 3′-splice-site variants in *DMD* and *TMC1* through 5′-splice-site strengthening. (**A**) The effect of the pathogenic variant c.1705-5T>G alone, the rescuing variant c.1812+4T>A alone, as well as their combination in *DMD* minigenes. Exon 15 skipping was almost completely rescued when c.1812+4T>A was introduced into the minigene. (**B**) Schematic overview of the *DMD* minigene containing Exons 14–16 used in A, annotated with the Human Splicing Finder (HSF) scores [0–100] of both variants. (**C**) Semi-quantification of the ratio of correctly (light gray rectangles) and aberrantly spliced (dark gray rectangles) RT-PCR products of WT splicing, as well as Exon 15 skipping in the *DMD* minigene. (**D**) The effect of the pathogenic variant c.237-6T>G alone, the rescuing variants c.362+4T>A and c.362+4T>G alone, and the combinations of the pathogenic variant with rescuing variants in the *TMC1* minigene. c.362+4T>A showed a slightly better splice defect rescue than c.362+6G>T. (**E**) Schematic overview of the *TMC1* minigene containing Exons 7 and 8 used in D, annotated with the HSF scores of all three variants. (**F**) Semi-quantification of the ratio of correctly (light gray rectangles) and aberrantly spliced (dark gray rectangles) RT-PCR products of WT splicing, as well as Exon 8 skipping in the *TMC1* minigene. When multiple aberrant products were observed, their percentages were summed up. * Band was identified as heteroduplex by Sanger sequencing. Red lettering denotes pathogenic variants. Green lettering indicates the variants that (partially) rescued exon skipping. ^#^ A fragment corresponding to a region on Chromosome 17 with high similarity to *TMC1*.

**Figure 3 ijms-21-02300-f003:**
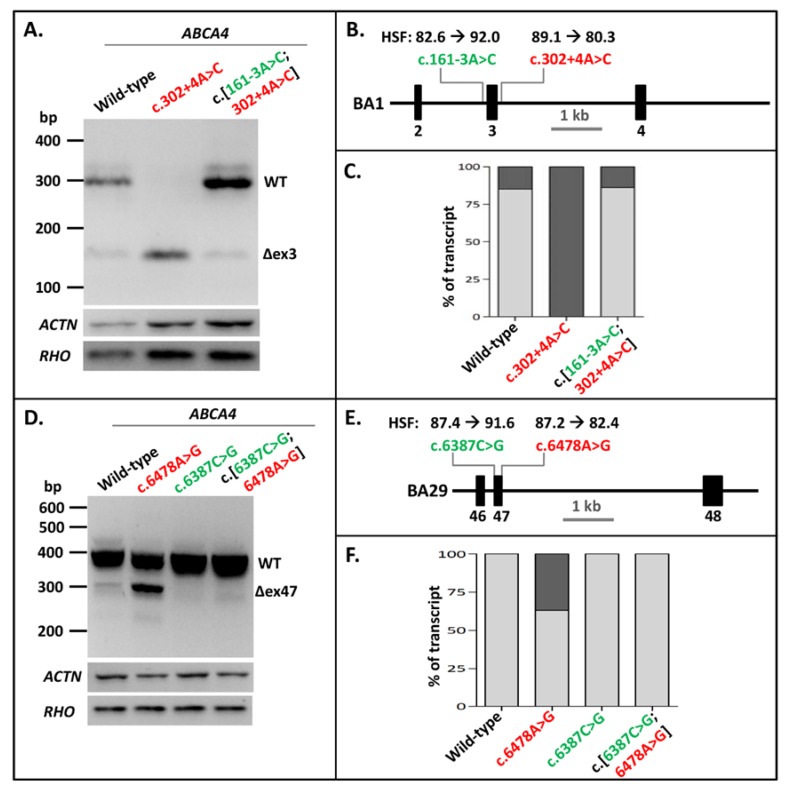
Strengthening 3′ splice sites rescues splice defects due to pathogenic 5′-splice-site variants in *ABCA4* Exons 3 and 47. (**A**) The effect of the pathogenic variant c.302+4A>C as well as the combination of this variant with the rescuing variant c.161-3A>C in construct BA1. The Exon 3 skipping was fully corrected. (**B**) Schematic overview of the *ABCA4* BA1 construct containing Exons 2–4 used in A, annotated with the Human Splicing Finder (HSF) scores [0–100] of both variants. (**C**) Semi-quantification of the ratio of correctly (light gray rectangles) and aberrantly spliced (dark gray rectangles) RT-PCR products of WT splicing, as well as Exon 3 skipping in the BA1 construct. (**D**) The effect of the pathogenic variant c.6478A>G, the rescuing variants c.6387C>T, and their combination in the BA29 construct. The partial Exon 47 skipping was fully rescued after the introduction of the 3′ splice-site variant. (**E**) Schematic overview of the BA29 construct containing Exons 46–48 used in D, annotated with the HSF scores of both variants. (**F**) Semi-quantification of the ratio of correctly and aberrantly spliced RT-PCR products of WT splicing as well as Exon 47 skipping in the BA29 construct. Red lettering indicates pathogenic variants. Green lettering indicates variants that rescued exon skipping. When multiple aberrant products were observed, their percentages were summed up.

**Figure 4 ijms-21-02300-f004:**
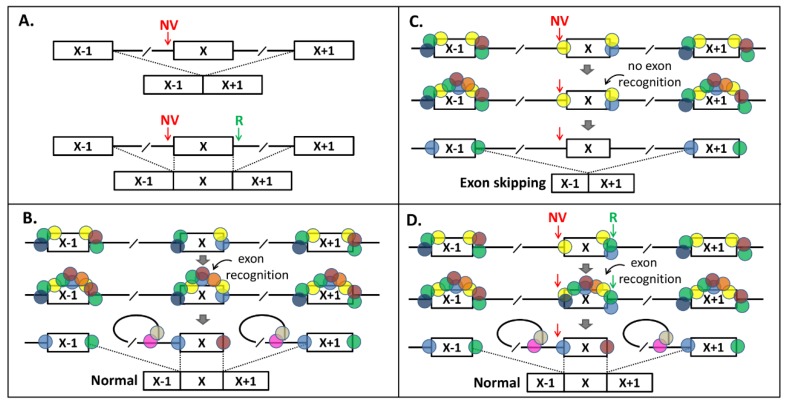
Splice-site-interdependency-based exon skipping rescue model. (**A**) Principle of rescue of Exon X skipping due to a noncanonical splice-site (NCSS) variant (NV) at the 3′ splice site (upper panel) by strengthening the 5′ splice site with a “rescue variant” (R, lower panel). (**B**) In normal splicing of Exon X, upon binding of splice junction sequence motifs by splice factors, additional splice factors will bind and define the exon, which is followed by the lariat configuration and normal splicing. Absence of one or more splice-enhancing motifs in Exon X makes this exon vulnerable to exon skipping when a NV variant is present, because the number of splice factors binding to Exon X is only just enough for exon definition. (**C**) In the presence of a NV in the 3′ splice site, insufficient splice factors will bind and there is no exon definition. (**D**) By strengthening the 5′ splice site with the “R”, additional splice proteins can bind and the exon definition and normal splicing are restored.

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
