# Peer review of "In or Out? New Insights on Exon Recognition through Splice-Site Interdependency"

_ijms, 2020, doi:10.3390/ijms21072300_

Round 1
Reviewer 1 Report
Dear authors you did not respond to any of the suggestions I've gave you.
The only changes I've found in the manuscript are "5' splice site" instead of SDS and "3'splice site" instead of SAS.
For such reason I do not recommend the publication of your manuscript.
Author Response
Reviewer 1:
Comments and Suggestions for Authors
Dear authors you did not respond to any of the suggestions I've gave you.
The only changes I've found in the manuscript are "5' splice site" instead of SDS and "3'splice site" instead of SAS.
Au: As listed below from our previous rebuttal, and as commented by Reviewer 2 (“the authors fulfilled most of them”), we did answer all questions to the best of our ability. Our only doubt was whether we should include the bar graphs that showed the quantification of our results either in the main figures or in the supplementals. Although no details on the issues that we did not address are provided by Reviewer 1, we have now included these bar graphs in Figures 1, 2 and 3 instead of the Supplemental data. We decided to put the sequence traces in Supplemental data as they were just meant to confirm what we expected, i.e. exon skipping events.
Below are our previous rebuttal points. As indicated by Reviewer 2, we addressed most of the points. We insert extra comments underlined, when needed.
Comment 1:
However it is not clear if patients with such combinations of splice and rescue variants have ever been described in clinics.
Au: Details of the pathogenic variants’ selection for ABCA4, DMD and TMC1 have been described in materials and methods section ‘’4.1. Noncanonical Splice Site Variant Selection’’. Rescue variants selected at the alternative splice sites were chosen based on the strength of their predicted effect, to show that a small increase in strength of a given splice site can compensate for the effect of a mild-moderate mutation on the other splice site of the same exon. Combinations of the selected pathogenic and rescue variants have not been reported in literature, as they would not result in disease.
Reviewer 1 (formerly: reviewer 2):
Cremers et al. showed reprogrammed splicing profiles of the disease-related alternative splicing (AS) events compared to those of the derived mutant splicing reporters based on in vitro splicing analysis. This work indeed increases the knowledge concerning splicing mechanism and impact on tissue development or inherited diseases. However, the manuscript contains many imprecisions and some of data analyses and drawn conclusions are wrong. The manuscript requires substantial reorganization.
Comment 1:
The level of novelty is also limited. Many conclusions were previously described in the literature. However, several analytic results make it difficult to interpret the conclusions as presented. Nevertheless, the novelty of this manuscript is limited to the knowledge of splicing regulation.
Au: As we listed in the introduction, very few data were published on the relationship between splice sites of the same exon, and the textbook dogma is that introns are recognized first by the splicing machinery. In this study we significantly expanded the knowledge on the important concept that exon-definition is very important.
It is not clear from the suggestions of the reviewer which conclusions are wrong. We can only repeat this question: can the reviewer be more specific?
Comment 2:
The percentage of splicing index (PSI) of each in vitro splicing assays shown in Figure 1, 2, and 3 should be presented in bar graph to clearly illustrate the reprogramming of splicing profiles. For example, the PSI of spliced products generated from WT ABCA4 or derived mutant reporter might be close as expected.
Au: Next to the gel pictures of the in vitro splicing assays shown in Figures 1, 2, and 3, we added bar graphs based on the percentages of normal and mutant splicing products due to the wildtype, mutant and combination of mutant and rescue variants in Supplementary Figure S1. We now added these data in Figures 1, 2 and 3.
Comment 3:
In Figure 1, the splicing profiles of WT ABCA4 and the derived mutants are not relevant to the HSF score of SDS or SAS within ABCA4 minigenes. In contrast, the splicing profiles of WT DMD, TMC1, or the derived mutants are relevant to the HSF score of SDS or SAS. These results should be further explained or discussed or the conclusions might be overestimated.
Au: The reviewer is correct that explanations are lacking for the results in Figure 1. In fact, in this part of the study we tested two hypothesis. First, as the +4 to +6 positions at the 5’ splice site of exon 39 constitute G’GT, alike the last nucleotide of an exon followed by two canonical splice site nucleotides, we tested whether changing the +4G to a C or T could already improve exon 39 inclusion although the HSF scores do not support this. Although we should keep in mind that the semi-quantification has intrinsic flaws as the small mutant PCR product will be preferentially amplified over the wild-type larger fragment and this is not the in vivo situation, there are no clear indications that the 5’ splice site is ‘weak’ because of this G’GT sequence at +4 to +6. Second, we tested the change from +4G>A as a rescue variant which also (by HSF) is predicted to increase this 5’ splice site strength and indeed rescued the splice defect induced by c.5461-10T>C. We inserted two paragraphs in the results to explain this part of the study:
“Based on Human Splicing Finder (HSF) the +4A variant strengthens this 5‘ splice site significantly (Figure 1B; from 83.8 to 92.1). The +4C and +4T variants, also according to HSF, do not strengthen this splice site, but were inserted to investigate another hypothesis. The +4 to +6 nucleotides are G’GT, which are part of a weak predicted 5’ splice site (HSF: 69.0) which could negatively influence proper exon 39 recognition. The +4G>C and +4G>T changes potentially could remove this effect and thereby result in less exon 39 skipping in the presence of the wild-type 3’ splice site sequence c.5461-10T or the mutant variant c.5641-10C.”
A few lines later: “As shown in Figure 1A and 1C, the +4G>C and +4G>T changes do not result in a clear rescue of erroneous splicing in the presence of the wild-type or mutant 3’ splice site, respectively, suggesting that the G’GT sequence at positions +4 to +6 does not significantly affect the strength of the 5’ splice site.”
On the contrary, in Figures 2 and 3, we only showed results of the wild-type construct, the mutant construct, and the construct with the mutant and rescue variants, each of which were expected to strengthen the splice site.
Can the reviewer outline what is missing after we provided this explanation?
In general, we are happy to make further adjustments to our manuscript if this improves its quality. It then would be appropriate if this reviewer can give more specific concerns.
Comment 4:
5'/3' splice site is better than splice donor/acceptor site for broad range of reader who is not familiar with splicing mechanism.
Au: We have replaced the terms splice donor/acceptor site with 5’/3’ splice site throughout the text.
Reviewer 2 Report
I totally agree with the previous reviewer’s concerns and the authors fulfilled most of them and corrected the manuscript accordingly. Only two minor point:
Why the recombinant cassettes are mentioned as midigenes? In the splicing field, they are generally named minigene/s. I would rename it for a broad audience. Some of the conclusions were already previously described. For example, in PMID: 29497141 authors are reporting as a natural change occurring within a mutated 5’ss is able to partially rescue the splicing defect. Moreover, in this article PMID: 27639833, authors reported that increasing the score of the 5’ss or 3’ss rescued the splicing defect caused by a synonymous change. These works strengthen the observation of the authors but should be mentioned and added in the discussion section.Author Response
Reviewer 2 (formerly reviewer 1):
Comments and Suggestions for Authors
I totally agree with the previous reviewer’s concerns and the authors fulfilled most of them and corrected the manuscript accordingly. Only two minor point:
Why the recombinant cassettes are mentioned as midigenes? In the splicing field, they are generally named minigene/s. I would rename it for a broad audience.
Au: We called them midigenes as most of them are much larger than the generally used minigenes that often only contain the exon of interest and minimal flanking sequences. In a few recent papers (Sangermano et al. Genome Research, 2018, PMID 29162642; Sangermano et al. Genetics in medicine PMID 30643219; Bauwens et al. Genetics in medicine, 2019, PMID 30670881;
Khan et al. Human mutation, 2019, PMID 31212395 and Fadaie et al. Human mutation, 2019, PMID 31397521) we have convincingly shown that the inclusion of flanking exons significantly increases the accuracy of the splice assay. Nevertheless, we understand this comment and have now replaced ‘midigene’ with ‘minigene’ throughout the manuscript.
Some of the conclusions were already previously described. For example, in PMID: 29497141 authors are reporting as a natural change occurring within a mutated 5’ss is able to partially rescue the splicing defect. Moreover, in this article PMID: 27639833, authors reported that increasing the score of the 5’ss or 3’ss rescued the splicing defect caused by a synonymous change. These works strengthen the observation of the authors but should be mentioned and added in the discussion section.
Au: We thank the reviewer for bringing these papers to our attention. The paper with PMID 2949714 does not deal with splice site interdependency but with a compensatory variant in the same splice site. Although it is of interest, we felt it did not contribute to discuss our data which fully deal with the effect of the two splice sites of the same exon. However, the results in PMID 27639833 are of interest, so we added a sentence at the end of the 3rd paragraph of the discussion: “Scalet et al. (2017) showed that strengthening of the 3’ or 5’ ss of ATR exon 9 could partially or fully rescue, respectively, the effect of an exonic variant that introduced a cryptic 5’ ss.”
Round 2
Reviewer 1 Report
I've read carefully the revised version of the manuscript and I've found the authors made substantial improvements and corrections according to the reviewer's suggestions.
I find the paper acceptable for publication but I recommend to the authors to redrawing figure 4, if possible, for a better understanding of the model they would like to propose.
This manuscript is a resubmission of an earlier submission. The following is a list of the peer review reports and author responses from that submission.
Round 1
Reviewer 1 Report
In this study the authors demonstrate that the so-called "splice site" interdependency represents a more general phenomenon for human inherited disease-associated variants and in particular they test the effect of 5 variants in non canonical ss sequences for ABCA4, DMD and TMC1 genes. The effect of the variant on uncorrected splicing could be rescued by strengthening the ss on the other side of the exon. Rescue variants were introduced and tested in an in vitro ss assay in HER293T cells.
This is an interesting paper suggesting new mechanism for splicing in human diseases.
The eventual therapeutic application of this findings is also very exciting.
However it is not clear if patients with such combinations of splice and rescue variants have ever been described in clinics.
Reviewer 2 Report
Cremer et al. showed reprogrammed splicing profiles of the disease-related alternative splicing (AS) events compared to those of the derived mutant splicing reporters based on in vitro splicing analysis. This work indead increases the knowledge concerning splicing mechanism and impact on tissue development or inherited diseases. However, the manuscript contains many imprecisions and some of data analyses and drawn conclusions are wrong. The manuscript requires substantial reorganization. The level of novelty is also limited. Many conclusions were previously described in the literature. However, several analytic results make it difficult to interpret the conclusions as presented. Nevertheless, the novelty of this manuscript is limited to the knowledge of splicing regulation.
Major Criticisms
The percentage of splicing index (PSI) of each in vitro splicing assays shown in Figure 1, 2, and 3 should be presented in bar graph to clearly illustrate the reprogramming of splicing profiles. For example, the PSI of spliced products generated from WT ABCA4 or derived mutant reporter might be close as expected. In Figure 1, the splicing profiles of WT ABCA4 and the derived mutants are not relevant to the HSF score of SDS or SAS within ABCA4 minigenes. In contrast, the splicing profiles of WT DMD, TMC1, or the derived mutants are relevant to the HSF score of SDS or SAS. These results should be further explained or discussed or the conclusions might be overestimated. 5'/3' splice site is better than splice donor/acceptor site for broad range of reader who is not familiar with splicing mechanism.